# Plasma Lipid Profiling of Three Types of Drug-Induced Liver Injury in Japanese Patients: A Preliminary Study

**DOI:** 10.3390/metabo10090355

**Published:** 2020-08-31

**Authors:** Kosuke Saito, Tatehiro Kagawa, Keiji Tsuji, Yuji Kumagai, Ken Sato, Shotaro Sakisaka, Naoya Sakamoto, Mitsuhiko Aiso, Shunji Hirose, Nami Mori, Rieko Tanaka, Toshio Uraoka, Kazuhide Takata, Koji Ogawa, Kazuhiko Mori, Motonobu Sato, Takayoshi Nishiya, Kazuhiko Takamatsu, Noriaki Arakawa, Takashi Izumi, Yasuo Ohno, Yoshiro Saito, Hajime Takikawa

**Affiliations:** 1Division of Medical Safety Science, National Institute of Health Sciences, Kanagawa 210-9501, Japan; saitok2@nihs.go.jp (K.S.); arakawa@nihs.go.jp (N.A.); 2Division of Gastroenterology and Hepatology, Department of Internal Medicine, Tokai University School of Medicine, Isehara 259-1193, Japan; kagawa@tokai.ac.jp (T.K.); s-hirose@is.icc.u-tokai.ac.jp (S.H.); 3Department of Gastroenterology, Hiroshima Red Cross Hospital and Atomic-Bomb Survivors Hospital, Hiroshima 730-8619, Japan; k-tsuji@hiroshima-med.jrc.or.jp (K.T.); morin@hiroshima-med.jrc.or.jp (N.M.); 4Kitasato University School of Medicine, 1-15-1 Kitasato, Minami-ku, Sagamihara, Kanagawa 252-0374, Japan; kuma-guy@za2.so-net.ne.jp (Y.K.); rieko_t@med.kitasato-u.ac.jp (R.T.); 5Department of Gastroenterology and Hepatology, Gunma University Graduate School of Medicine, Maebashi 371-8511, Japan; satoken@gunma-u.ac.jp (K.S.); uraoka@gunma-u.ac.jp (T.U.); 6Department of Gastroenterology, Fukuoka University Faculty of Medicine, Fukuoka 814-0180, Japan; sakisaka@fukuoka-u.ac.jp (S.S.); edihuzak_t@yahoo.co.jp (K.T.); 7Department of Gastroenterology and Hepatology, Hokkaido University Faculty of Medicine and Graduate School of Medicine, Sapporo 060-8648, Japan; sakamoto@med.hokudai.ac.jp (N.S.); k-ogawa@med.hokudai.ac.jp (K.O.); 8Department of Medicine, Teikyo University School of Medicine, Tokyo 173-8606, Japan; aiso@med.teikyo-u.ac.jp (M.A.); takikawa@med.teikyo-u.ac.jp (H.T.); 9Daiichi Sankyo Co., Ltd., Tokyo 134-8630, Japan; mori.kazuhiko.md@daiichisankyo.co.jp; 10Astellas Pharma Inc., Tsukuba 305-8585, Japan; motonobu.sato@astellas.com (M.S.); kazuhiko.takamatsu@astellas.com (K.T.); 11Daiichi Sankyo RD Novare Co., Ltd., Tokyo 134-8630, Japan; nishiya.takayoshi.ec@rdn.daiichisankyo.co.jp; 12Kihara Memorial Foundation, Yokohama 230-0045, Japan; t-izumi@kihara.or.jp (T.I.); yasuokunhe@ra3.so-net.ne.jp (Y.O.); 13Faculty of Medical Technology, Teikyo University, Tokyo 173-8606, Japan

**Keywords:** lipidomics, drug-induced liver injury, biomarker, plasma lipid profiles

## Abstract

Drug-induced liver injury (DILI) is a major adverse event caused by drug treatment, which can be categorized into three types: hepatocellular, mixed, and cholestatic. Although nearly every class of drugs can cause DILI, an overall understanding of lipid profiles in DILI patients is lacking. We used lipidomics to analyze the plasma lipid profiles of patients to understand their hepatic pathophysiology and identify DILI biomarkers. We identified 463 lipids and compared their levels between the acute and recovery phases of the three types of DILI patients. Mixed and cholestatic types demonstrated specific plasma lipid alterations between the phases, but the hepatocellular type did not. Moreover, as specific indicators of mixed-type DILI, levels of several ceramides increased in the acute phase, while those of arachidonic acid-containing ether-linked phosphoglycerolipids decreased. In contrast, as specific indicators of cholestatic-type DILI, levels of palmitic acid-containing saturated or monounsaturated phosphatidylcholines increased in the acute phase, while those of arachidonic acid- or docosahexaenoic acid-containing ether-linked phosphoglycerolipids and phosphatidylinositols decreased. We also identified lipids with a relatively high capacity to discriminate the acute phase from the recovery phase and healthy subjects. These findings may help with understanding the pathophysiology of different DILI types and identify candidate biomarkers.

## 1. Introduction

Drug-induced liver injury (DILI) is a major adverse event caused by drug treatment and is the most frequent cause of acute liver failure in the U.S. [1,2]. Depending on the histological location of the tissue damage, DILI is categorized as hepatocellular, cholestatic, or mixed type, which is usually based on changes in blood levels of alanine transaminase (ALT) and alkaline phosphatase (ALP). The causal relationship between DILI and suspected drugs has been digitized by the CIOMS/RUCAM and DDW-J2004 scoring scales (in Japan), which are used in clinical practice [3,4,5]. The mechanisms of DILI are diverse and include direct toxicity by the administered drug or its metabolites and immune reactions against the drug or its metabolites [6,7]. The most studied drug causing DILI is acetaminophen, which is metabolized to a toxic and electrophilic intermediate by cytochrome P450 isoenzymes (such as CYP2E1 and CYP3A4); this intermediate interacts with intracellular proteins resulting in hepatocyte damage [8]. Although specific mechanisms of drugs with relatively high incidence of DILI have also been studied [6,7], nearly every class of drug can cause DILI. However, biomarkers and characteristics of DILI that are important to understand its pathophysiology are limited.

Lipids, such as phosphoglycerolipids, sphingolipids, and neutral lipids, are components of cellular membranes that also play important roles in multiple biological processes, including apoptosis, inflammation, proliferation, and differentiation [9,10,11,12]. The liver is a central organ in regulating lipid levels, and therefore, aberrations in lipid homeostasis are associated with hepatic injury and disease. In addition, a recent study demonstrated that the composition of plasma lipids correlates well with that of hepatic lipids [13]. Thus, plasma lipid profiles could be useful tools to understand the biological processes in the liver. To analyze plasma lipid profiles, lipidomics based on mass spectrometry has been established [14,15,16,17]. Plasma lipidomics has already been used to study hepatocellular carcinoma [18,19], liver phospholipidosis [20], nonalcoholic fatty liver disease [21], and other hepatic diseases and toxicities. For example, plasma lipidomics of hepatocellular carcinoma demonstrated decreased levels of lysophosphatidylcholine (LPC) in plasma, suggesting the hepatic activation of autotoxin and its involvement in hepatocarcinogenesis [22]. Moreover, plasma lipidomics of liver phospholipidosis demonstrated increased levels of d18:1/24:0 glucosylceramide (GluCer), which was proposed as a biomarker for the disease [20]. Therefore, the characterization of overall plasma lipid profiles could lead to a better understanding of hepatic pathophysiology and identify new DILI biomarkers.

In this study, we aimed to analyze the differences in lipid profiles among three DILI types (hepatocellular, mixed, and cholestatic) during acute and recovery phases in human patients. We present novel lipidomic data for the three injury types, which could be used to screen for DILI biomarkers and/or develop future novel therapies by understanding lipid homeostasis in DILI.

## 2. Results

### 2.1. DILI Patients Recruited in the Present Study

We recruited 54 DILI patients, comprising 33 hepatocellular, 13 mixed, and 8 cholestatic types (Table 1). Of these patients, 11 males and 22 females were diagnosed with hepatocellular type, 9 males and 4 females were diagnosed with mixed type, and 4 males and 4 females were diagnosed with cholestatic type. Their median ages were 56, 60, and 69 for the hepatocellular, mixed, and cholestatic types, respectively. The median CIOMS/RUCAM scores were eight, nine, and eight for the hepatocellular, mixed, and cholestatic types, respectively. In addition, the median DDW-J2004 scores were eight for each of the respective DILI patient types. The causal relationship between suspected drug and liver damage was definite in all patients using the CIOMS/RUCAM scale and all patients using the DDW-J 2004 score, except for one case.

The suspected drug with the highest frequency of culpability was loxoprofen, which was responsible for four cases out of the 54 patients (two, one, and one case in the hepatocellular-, mixed-, and cholestatic-type patients, respectively). In addition, when the prescribed drugs were categorized according to the World Health Organization (WHO) Anatomical Therapeutic Chemical (ATC) codes, the highest number of cases was found in antibacterial agents for systemic use (J01, eight cases), followed by antineoplastic agents (L01, six cases), anti-inflammatory and antirheumatic products (M01, six cases), and psycholeptics (N05, six cases). The causes of DILI were widely diverse among cases, which hindered the analysis of drug-specific or drug category-specific effects.

### 2.2. Global Plasma Lipid Profiling in the Three DILI Types

Global plasma lipidomic profiling using our lipidomics platform detected 463 lipids spanning 31 lipid classes (Appendix A and summarized in Table 2). Note that in our assay platform, unconjugated bile acids were detectable but not quantitative because their liquid chromatography (LC) retention time is close to the void fraction where ionization is unstable due to the presence of unretained salts. The exemplar LC/MS traces are shown in Appendix A. To distinguish stereoisomers, each quantified lipid was assigned a specific metabolite ID. The fatty acid side chains in the lipids were confirmed using mass spectrometry (MS), the confirmed fatty acid fragments were indicated after a semicolon in the name. The combination of fatty acid side chains was combined using a slash. If two different sets of fragments were confirmed, we provided both of them, separated by a comma. The identified lipids comprised 184 phospholipids, 80 sphingolipids, 180 neutral lipids, and 19 others, including coenzyme Q10 (CoQ10), free fatty acids (FAs), and acylcarnitines (Cars). The major phospholipid class, phosphatidylcholines (PCs), contained 56 lipids. In addition, the major sphingolipid class, sphingomyelins (SMs), comprised 37 lipids, and the major neutral lipid class, triacylglycerols (TGs), comprised 138 lipids. The identified lipid levels were compared between the acute and recovery phases in each DILI type. Lipids with both high effect sizes (g > 0.8) and statistically significant differences (*p* < 0.05) were defined as altered.

Although 112 lipids were significantly different between the phases, no lipid was defined as altered in the hepatocellular type (Figure 1a). In contrast, 9 and 20 lipids were defined as altered in the mixed and cholestatic types, respectively (Figure 1b,c). Thus, we focused on the mixed and cholestatic types for further analysis.

### 2.3. Discrimination Ability for Mixed-Type DILI between Acute Phase and Recovery Phase or Healthy Volunteers

In the mixed-type DILI patients, three lipids, ceramide (Cer)(d34:1; d18:1/16:0), Cer(d36:1; d18:1/18:0), and oxidized ganglioside GM3 (GM3+O)(d34:1), were increased in the acute phase compared with the recovery phase, while six lipids, LPC(18:2), ether-linked LPC LPC(16:1e), ether-linked PC PC(38:6e; 18:2e/20:4, 16:1e/22:5), ether-linked phosphatidylethanolamine (PE) PE(36:4e; 16:0e/20:4), PE(38:4e; 18:0e/20:4), and PE(38:6e; 18:2e/20:4), were decreased in plasma (Table 3). The increased lipid of the highest effect size in the mixed-type patients was Cer(d34:1; d18:1/16:0), and the corresponding decreased lipid was PE(38:4e; 18:0e/20:4). All PCes and PEes contained the same FA (20:4: arachidonic acid).

Once we had characterized the specific lipids that were altered in mixed-type DILI, we next evaluated their discrimination ability between the acute phase and recovery phase by receiver operating characteristics (ROC) analysis. As shown in Table 3, four lipids, PE(38:4e; 18:0e/20:4), Cer(d34:1; d18:1/16:0), Cer(d36:1; d18:1/18:0), and GM3(d34:1)+O, had area under the curve (AUC) values over 0.8. The lipid with the highest AUC was Cer(d34:1; d18:1/16:0), with a value of 0.87.

We further compared the lipids levels of acute phase mixed-type DILI patients with the lipid levels of healthy subjects. Although the median ages of the three DILI patient types were approximately 60 years, we recruited the healthy subjects in four groups according to sex and age (HM1; middle-age male, HM2; old-age male, HF1; middle-age female, HF2; old-age female, where middle age was approximately 45 years and old age was approximately 60 years) (Appendix A). The different lipids between mixed or cholestasis type DILI and all healthy subjects were listed in Appendix A (mixed) and Appendix A (cholestasis). As shown in Table 3, 6 lipids, LPC(18:2), LPC(16:1e), PE(38:6e; 18:2e/20:4), Cer(d34:1; d18:1/16:0), Cer(d36:1; d18:1/18:0), and GM3(d34:1)+O, were significantly different when comparing the acute phase DILI patients with all groups of healthy subjects. Cer(d34:1; d18:1/16:0), Cer(d36:1; d18:1/18:0), and GM3(d34:1)+O also had AUC values > 0.8 by ROC analysis versus all groups of healthy subjects. The representative individual plots of lipid levels discriminating the acute phase of mixed-type DILI from the recovery phase or the healthy volunteer groups are shown in Figure 2. Furthermore, we also calculated the ratio of altered specific lipids and evaluated their discriminating ability to acute phase mixed-type DILI patients from other groups, but no ratio of altered specific lipids further improved the discriminating ability. In addition, the absolute correlation coefficient of the altered specific lipids in mixed-type DILI with clinical parameters (AST, ALT, ALP, and total bilirubin) were all less than 0.6 (Appendix A).

### 2.4. Discrimination Ability for Cholestatic-Type DILI between Acute Phase and Recovery Phase or Healthy Volunteers

In the cholestatic-type DILI patients, 4 lipids, PC(30:0; 14:0/16:0), PC(31:0; 15:0/16:0), PC(32:1; 16:0/16:1), and PC (33:1; 15:0/18:1, 16:0/17:1), were increased in the acute phase then the recovery phase, while 16 lipids, PC(36:5e; 16:1e/20:4), PC(38:6e; 18:2e/20:4, 16:1e/22:5), PE(36:4e; 16:0e/20:4), PE(38:4e; 18:0e/20:4), PE(40:6e; 18:0e/22:6), PE(40:7e; 18:1e/22:6; M160), PE(40:7e; 18:1e/22:6; M161), phosphatidylinositol (PI)(38:3), PI(38:4; 18:0/20:4), PI(40:4), triglycosylceramide (CerG3)(d40:1), CerG3(d42:1), CerG3(d42:2), SM(d40:1; d18:1/22:0), TG(44:0; 14:0/14:0/16:0, 12:0/16:0/16:0), and CoQ10, were decreased (Table 4). The increased lipid of highest effect size in the cholestatic-type patients was PC(33:1; 15:0/18:1, 16:0/17:1) and the corresponding decreased lipid was PE(40:7e; 18:1e/22:6). Three FA(20:4)-containing ether-linked phospholipids, PC(38:6e; 18:2e/20:4, 16:1e/22:5), PE(36:4e; 16:0e/20:4), and PE(38:4e; 18:0e/20:4), were common with the mixed-type cases, but FA(22:6), corresponding to docosahexaenoic acid, was contained in three PEes, PE(40:6e; 18:0e/22:6), PE(40:7e; 18:1e/22:6; M160), and PE(40:7e; 18:1e/22:6; M161), which are specific for the cholestatic-type cases. In addition, all increased PCs in the acute phase of cholestatic-type patients contained the same FA(16:0: palmitic acid).

We also evaluated the discrimination ability of specific lipids that were altered in cholestatic-type DILI between the acute and recovery phases by ROC analysis. As shown in Table 4, 12 lipids, PC(31:0; 15:0/16:0), PC (33:1; 15:0/18:1, 16:0/17:1), PE(36:4e; 16:0e/20:4), PE(38:4e; 18:0e/20:4), PE(40:6e; 18:0e/22:6), PE(40:7e; 18:1e/22:6; M160), PI(38:3), PI(38:4; 18:0/20:4), PI(40:4), CerG3(d40:1), CerG3(d42:1), and CoQ10, had AUC values over 0.8. The lipid with the highest AUC was PE(40:7e; 18:1e/22:6; M160), with a value of 0.91.

We further compared the lipids levels of acute phase cholestatic-type DILI patients with the lipid levels of healthy subjects (grouped as indicated in Section 2.3). As shown in Table 4, eight lipids, PC(30:0; 14:0/16:0), PC(31:0; 15:0/16:0), PC(32:1; 16:0/16:1), PC(33:1; 15:0/18:1, 16:0/17:1), PC(36:5e; 16:1e/20:4), PI(38:3), PI(38:4; 18:0/20:4), and SM(d40:1; d18:1/22:0), were significantly different when comparing the acute phase DILI patients with all the compared groups of healthy subjects. PC(30:0; 14:0/16:0), PC(31:0; 15:0/16:0), PC(32:1; 16:0/16:1), PC (33:1; 15:0/18:1, 16:0/17:1), PI(38:3), PI(38:4; 18:0/20:4), and SM(d40:1; d18:1/22:0) also had AUC values >0.8 using ROC analysis versus all the groups of healthy subjects. The representative individual plots of lipid levels discriminating cholestatic-type DILI in the acute phase from the recovery phase or the healthy volunteer groups are shown in Figure 3. Furthermore, we also calculated the ratio of altered specific lipids and evaluated their discriminating ability to acute phase cholestatic-type DILI patients from other groups, but no ratio of altered specific lipids further improved the discriminating ability. In addition, the correlation coefficient of the altered specific lipids in mixed-type DILI with clinical parameters (AST, ALT, ALP, and total bilirubin) demonstrated over 0.6 (with *p* < 0.05) for three out of four palmitic acid-containing saturated or monounsaturated PCs, PC(31:0; 15:0/16:0), PC(32:1; 16:0/16:1), and PC (33:1; 15:0/18:1, 16:0/17:1) (Appendix A). The absolute correlation coefficient of all other specific lipids was less than 0.6.

## 3. Discussion

In this study, we used plasma lipid profiling to characterize the pathophysiology of three different types of DILI in human patients and made five broad observations. First, the mixed and cholestatic types of DILI demonstrated specific plasma lipid alterations between acute and recovered phases, but the hepatocellular type did not. Second, as specific features of mixed-type DILI, when compared with levels in the recovery phase, several ceramides were increased in the acute phase, while arachidonic acid-containing ether-linked phosphoglycerolipids were decreased. Third, as specific features of cholestatic-type DILI, when compared with levels in the recovery phase, palmitic acid-containing saturated or monounsaturated PCs increased in the acute phase, while arachidonic acid- or docosahexaenoic acid-containing ether-linked phosphoglycerolipids and PIs decreased. Fourth, of the specific lipids altered in mixed-type DILI, the levels of Cer(d34:1; d18:1/16:0), Cer(d36:1; d18:1/18:0), and GM3(d34:1)+O demonstrated relatively high discrimination ability for the acute phase over the recovery phase in all groups of healthy subjects. Finally, of the specific lipids altered in cholestatic-type DILI, the levels of PC(31:0; 15:0/16:0), PC(33:1; 15:0/18:1, 16:0/17:1), PI(38:3), and PI(38:4; 18:0/20:4) demonstrated relatively high discrimination ability for the acute phase over the recovery phase in all groups of healthy subjects.

Although the number of subjects in the cholestatic-type DILI group was limited, the number of specific lipids altered was larger in this group than in the other two groups. This result suggests that the alteration in hepatic lipid homeostasis in cholestatic-type DILI shares a common mechanism among diverse suspected drugs. One representative plasma lipid that was increased in cholestatic-type DILI was palmitic acid (16:0)-containing saturated or monounsaturated PCs. Palmitic acid has been reported as the major fatty acid in biliary PCs [23]. In addition, the partner FAs of palmitic acid in the specifically altered PCs in cholestatic-type DILI, FA(16:1), and FA(17:1) are preferentially secreted into bile [24]. Thus, these increased levels of palmitic acid-containing saturated or monounsaturated PCs in the plasma were probably due to the reduced bile secretion of palmitic acid-containing saturated or monounsaturated PCs by cholestasis. This is also supported because the total bilirubin levels, which were also elevated by biliary structure, were highly correlated with the lipid levels in the DILI patients in this study.

Along with the palmitic acid-containing saturated or monounsaturated PCs, PIs, such as PI(38:3) and PI(38:4; 18:0/20:4), were also specifically increased in the cholestatic-type DILI patients. To date, the role of increased plasma PIs in cholestatic-type DILI remains unclear. However, supplementation with PIs decreases mRNA levels of the inflammatory cytokines/chemokines, tumor necrosis factor-alpha (TNF-α), and monocyte chemoattractant protein-1 (MCP-1), which are upregulated in steatosis [25]. In addition, blood and liver PIs were shown to increase with hepatic steatosis [26,27]. Therefore, one plausible reason for the increased PIs in plasma is to counteract the hepatic inflammation that can occur with lipid dysregulation.

Unlike other lipid classes, arachidonic acid-containing ether-linked phosphoglycerolipids were commonly altered in mixed and cholestatic-type DILI. Decreased levels of serum ether-linked phosphoglycerolipids have been reported in patients with nonalcoholic steatohepatitis and nonalcoholic fatty liver disease when compared to the levels in healthy controls [21]. In addition, plasma and liver ether-linked phosphoglycerolipid levels were decreased in a valproic acid-induced rat model of hepatic steatosis [28]. Thus, the decreased levels of ether-linked phosphoglycerolipids that we observed in the plasma of mixed and cholestatic-type DILI patients could be caused by mechanisms that are like those in steatosis and steatohepatitis, and they could reflect reduced levels in the liver. Arachidonic acid is well-known to be metabolized to inflammatory eicosanoids, such as prostaglandin E_2_; thus, decreased levels of arachidonic acid-containing ether-linked phosphoglycerolipids in the plasma and the liver in the reference would implicate the inflammatory incidences in the liver of mixed and cholestatic-type DILI patients as well as patients with steatosis and steatohepatitis. Alternatively, ether-linked phosphoglycerolipids have been characterized as peroxisome-synthesized lipids and are a key component of peroxisome [29]. In fact, decreased levels of hepatic glyceronephosphate O-acyltransferase, which is a key peroxisomal enzyme for the synthesis of ether-linked phosphoglycerolipids, have been observed in a rat model of hepatic steatosis [28]. In addition, rescuing ether-linked phosphoglycerolipid levels by alkyl glycerol treatment could prevent impaired peroxisomal metabolism and hepatic steatosis [30,31]. Taken together, the decreased levels of plasma ether-linked phosphoglycerolipids that we observed may be caused by peroxisomal dysfunction in mixed and cholestatic types of DILI, and the rescue of ether-linked phosphoglycerolipid levels could be utilized for the therapeutic treatment of these DILI types.

Besides arachidonic acid-containing phosphoglycerolipids, increased plasma Cer was a characteristic feature of mixed-type DILI. To date, whether the increase in Cers plays a pivotal role in mixed-type DILI is unclear. However, Cers possess cell-signaling properties that are relevant to inflammation and apoptosis [32,33], and they may be involved in cystic fibrosis in the lung [34,35]. Thus, it is reasonable to speculate that increased Cer levels in mixed-type DILI patients contribute to hepatic inflammation and trigger subsequent pathological fibrosis.

In the present study, we also evaluated the differences in plasma lipids and their ability to discriminate between acute state DILI and healthy subjects divided into four age/sex groups. We identified 3 and 4 lipids in mixed and cholestatic types of DILI, respectively, as lipids with high discrimination ability. Although their scores did not exceed those of ALT and ALP (data not shown), these lipids could be utilized as biomarkers for DILI patients with ALT and ALP levels that are not diagnostic of liver disease. For example, ALT is elevated in patients with muscle injury and ALP is elevated in bone diseases. These lipids may also be helpful to discriminate DILI types and determine therapeutic approaches. Further analysis is needed to corroborate these speculations.

There are several limitations in the present study. First, it was performed with a few subjects of mixed and cholestatic types. Although we collected samples in both the acute and recovery phases from the same patients, the number of analyzed patients was limited, thus restricting the statistical power of our analysis. Second, due to the sparse number of events and limited ability to follow up patients, we recruited DILI patients from seven core hospitals. Although we used the same sampling protocol, hospital-to-hospital variation in sample preparation may have produced slightly different results in plasma lipid levels. Third, postprandial effect has been reported to have a global impact on lipidomics, although the impact is less than that of inter-individual variations [36,37]. Thus, this impact should be taken into consideration even though it is less than the impact caused by inter-individual variations. However, it is difficult to control the food intake of DILI patients, especially during the acute phase. Therefore, we believe that the state of fasting can be disregarded for this preliminary study. Fourth, although we recruited self-reported healthy subjects who had taken no medication for at least 1 week as controls, they may have been unaware of their disease status. Fifth, we did not control the alcohol and food intake of the patients, and the time of blood draw was not standardized, both of which might have affected plasma lipid levels. Sixth, since we used one internal standard (PC[1 2:0/12:0]) for all the classes of lipids, ionization efficiency should be different among the classes. Thus, the fold changes can effectively be calculated/estimated even using the same IS for all the lipids, but the comparison between lipid classes is not valid then. Last, it is difficult to consider the effects of other disease states and external factors, such as sexes and ages. In fact, several lipids, such as PEes and CerG3s, have high discriminant ability between acute phase and recovered phase DILI, while those lipids could not discriminate acute phase DILI and some groups of healthy subjects, which may be attributed to differences in sexes and ages. In addition, as was reported in the literature, many diseases, including liver-related diseases, which are possibly base diseases and complications, alter the plasma lipid levels [18,19,20,21,22]. Multivariate analysis including these potentially affecting factors should be performed using more patients’ samples. Therefore, to address these limitations, a future, large-scale study with updated protocols should be performed.

In conclusion, we characterized the plasma lipid profiles of three types of DILI patients using a lipidomics approach. By comparing samples in acute and recovery phases, we revealed that mixed and cholestatic types of DILI produce specific alterations in plasma lipid profiles. In addition, by comparing these data to those of healthy subjects, we found several candidate markers of mixed and cholestatic DILI that discriminate the acute phase from the recovery phase and healthy state. Our study provides insights into the alterations in plasma lipidomic profiles, which reflect alterations in lipid homeostasis in the livers of DILI patients. These findings may help to understand the pathophysiology of different types of DILI.

## 4. Materials and Methods

### 4.1. Subjects and Sample Collection

DILI patients were recruited at the Teikyo University Hospital, Tokai University Hospital, Hiroshima Atomic-bomb Survivors Hospital, Kitasato University Hospital, Gunma University Hospital, Fukuoka University Hospital, and Hokkaido University Hospital. The inclusion criteria for DILI in the acute phase were ALT ≥150 U/L and/or ALP ≥ 2× upper limit of normal, as described previously [38,39]. In addition, each DILI patient was scored using the CIOMS/RUCAM [3] and DDW-J2004 [4,5] scales, and the highest probability cases in these scores were included in this study. The CIOMS/RUCAM scale involves a scoring system that categorizes the cases into “definite or highly probable” (score > 8), “probable” (score 6–8), “possible” (score 3–5), “unlikely” (score 1–2), and “excluded” (score ≤ 0). The DDW-J2004 scale involves a scoring system that categorizes the cases into “highly probable” (score > 5), “possible” (score 3–4), and “unlikely” (score ≤ 2). The DILI type and entry into the recovery phase were also diagnosed by DILI experts at each hospital. All healthy subjects were non-smoking, self-reported healthy volunteers who had taken no medications for at least 1 week before the study.

Blood samples were collected by venipuncture into 7 mL EDTA-2Na-containing vacuum blood collection tubes (VENOJECT II, TERUMO, Tokyo, Japan). The blood samples were immediately centrifuged (2500× *g*, 10 min, 4 °C); the resulting plasma was dispensed into screw-capped polypropylene tubes and stored in a deep freezer (−80 °C) before use. The plasma was typically frozen within 2 h from blood draw, although this occasionally extended to 4 h.

This study was conducted in accordance with the Declaration of Helsinki and approved by the Ethics Committee of the National Institute of Health Science (256, and 260 for Kihara Memorial Foundation), Teikyo University Hospital (15-127-2), Tokai University Hospital (15R-117), Hiroshima Atomic-bomb Survivors Hospital (H27-399-2), Kitasato University Hospital (B13-182), Gunma University Hospital (1487), Fukuoka University Hospital (18-8-04), Hokkaido University Hospital (016-0345), Daiichi Sankyo Co., Ltd. (15-0504-00), and Astellas Pharma Inc. (150028-01, 150047-01). Written informed consent was obtained from all participants.

### 4.2. Lipidomics

Lipid extraction was performed using the Microlab NIMBUS workstation (Hamilton, Binaduz, GR, Switzerland). The plasma samples were mixed with nine volumes of methanol/isopropanol (1/1) containing an internal standard (PC[12:0/12:0]), which is not detectable endogenously, at 2 μM. The mixed samples were filtered through a FastRemover Protein Removal Plate (GL Science, Tokyo, Japan) using an MPE2 automated liquid handling unit (Hamilton). The resulting lipid-containing filtrate was directly subjected to lipidomics. To obtain the lipidomics data, we performed reversed-phase LC (RPLC; Ultimate 3000, Thermo Fisher Scientific, Waltham, MA, USA) and MS (Orbitrap Fusion, Thermo Fisher Scientific), as described previously [40,41]. Compound Discoverer 2.1 (Thermo Fisher Scientific) was used with the raw data for peak extraction, annotation, identification, and lipid quantification, as described previously with a prior version of the software [40,41]. For isomers (same class, carbon length, and number of double bonds) showing different retention times in RPLC, each lipid was assigned a metabolite ID to distinguish it. Lipids with two different fatty acid combinations (e.g., 38:6e; 18:2e/20:4, 16:1e/22:5) indicate that the quantified lipid is a mixture of two different lipids that could not be separated. The quantified raw data were normalized to the internal standard. Since the lipidomics analysis was combined across two batches, the median value of each lipid in all samples was set to one in each batch to consolidate data from two batches after normalization. The processed data for the lipid levels are presented in Appendix A.

### 4.3. Statistical Analysis

Significant differences in lipid levels were assessed by paired t-tests and Welch’s t-test, and the effect size, which is calculated by Hedge’s *g*, was considered. In this study, due to the limitation of sample size, a lipid level was considered specifically altered if its *p* value was <0.05 and its absolute effect size was >0.8. The discrimination ability was assessed by AUC score in ROC analysis using GraphPad Prism 6 (GraphPad Software, San Diego, CA, USA). The correlation coefficient was calculated as Pearson’s correlation coefficient.

## Figures and Tables

**Figure 1 metabolites-10-00355-f001:**
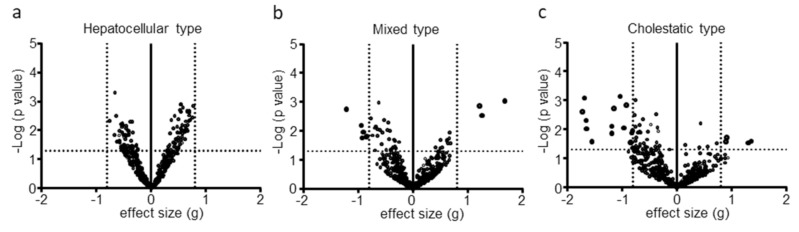
Volcano plot of lipid alterations in three types of DILI. Statistical probability (*p* value) and effect size (g) were determined by a comparison of lipid levels between acute phase and recovery phase of the DILI patients. Volcano plots show -log *p* value versus g value for (**a**) hepatocellular type, (**b**) mixed type, and (**c**) cholestatic type. Each dot represents an individual lipid.

**Figure 2 metabolites-10-00355-f002:**
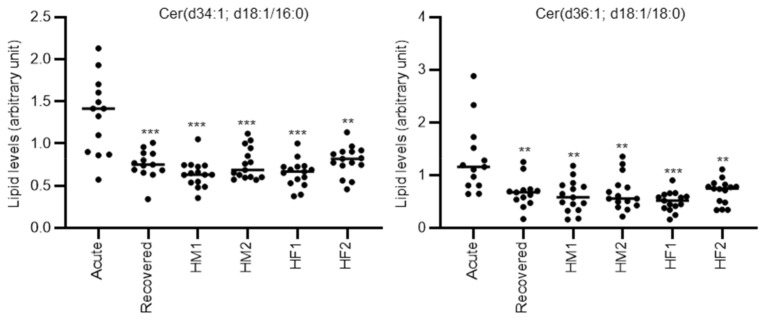
Representative individual plots of lipids levels discriminating the acute phase of mixed-type DILI from the recovery phase and the four healthy volunteer groups. Each dot represents an individual sample. Statistical significance is indicated as follows: ** *p* < 0.01, *** *p* < 0.001. Acute; acute phase DILI patients, Recovered; recovery phase DILI patients, HM1; healthy male subject group 1 (approximately 45 years old), HM2; healthy male subject group 2 (approximately 60 years old), HF1; healthy female subject group 1 (approximately 45 years old), HF2; healthy female subject group 2 (approximately 60 years old).

**Figure 3 metabolites-10-00355-f003:**
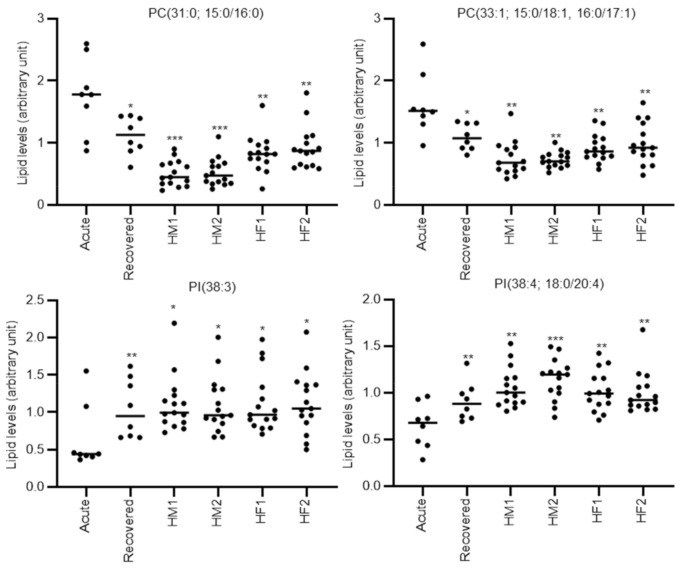
Representative individual plots of lipids levels discriminating the acute phase of cholestatic-type DILI from the recovery phase and the four healthy volunteer groups. Each dot represents an individual sample. Statistical significance is indicated as follows: * *p* < 0.05, ** *p* < 0.01, *** *p* < 0.001. Acute; acute phase DILI patients, Recovered; recovery phase DILI patients, HM1; healthy male subject group 1 (approximately 45 years old), HM2; healthy male subject group 2 (approximately 60 years old), HF1; healthy female subject group 1 (approximately 45 years old), HF2; healthy female subject group 2 (approximately 60 years old).

**Table 1 metabolites-10-00355-t001:** Clinical information of patients in this study.

DILI Type	Hepatocellular	Mixed	Cholestatic
no. of subjects	33	13	8
CIOMS/RUCAM scale; median (quartile)	8 (7–9)	9 (7–9)	8 (7.5–9)
DDW-J 2004 score; median (quartile)	8 (7–9)	8 (7–8)	8 (7–8.5)
Sex; male/female	11/22	9/4	4/4
Age; median (quartile)	56 (46–68)	60 (57–76)	69 (64.5–72.5)
BMI; median (quartile)	22.7 (19.8–24.1)	22.1 (21.3–23.3)	25.3 (22.8–26.1)
acute phase AST(U/L); median (quartile)	239 (102–526)	130 (96–191)	81 (55.75–200.5)
acute phase ALT(U/L); median (quartile)	336 (204–963)	196 (161–423)	97 (84.75–131)
acute phase ALP(U/L); median (quartile)	360 (277–410)	555 (448–914)	1465 (1129.5–1721.5)
acute phase T. Bil(mg/dl); median (quartile)	1 (0.7–2.2)	0.6 (0.5–1.3)	1.5 (0.8–3.65)
recovered phase AST(U/L); median (quartile)	23 (19–29)	31 (20–34)	28.5 (19.5–30.25)
recovered phase ALT(U/L); median (quartile)	25 (17–35)	32 (23–48)	22 (15–30.25)
recovered phase ALP(U/L); median (quartile)	249 (185.75–324.75)	316 (244–373)	266.5 (190.25–332)
recovered phase T. Bil(mg/dl); median (quartile)	0.7 (0.525–0.975)	0.65 (0.5–0.925)	0.7 (0.6–0.9)
Cause			
-Prescribed drugs	26	10	6
-Other	1	1	0
-Undefined	6	2	2
Suspected drugs; ad. in over 2 DILI patients			
-Acetaminophen	2	1	0
-Cefditoren	2	0	0
-Cyclophosphamide	2	0	0
-Febuxostat	0	1	1
-Gemcitabine	2	0	0
-Loxoprofen	2	1	1
-Nifedipine	1	0	1
ATC level 2 of suspected drugs; ad. in over 3 DILI patients (ATC code in parenthesis)			
-calcium channel blockers (C08)	2	0	1
-antibacterials for systemic use (J01)	6	2	0
-antineoplastic agents (L01)	5	1	0
-anti-inflammatory and antirheumatic products (M01)	3	2	1
-psycholeptics (N05)	4	1	1

AST; aspartate transaminase, ALT; alanine transaminase, ALP; alkaline phosphatase, T. Bil; total bilirubin, ad.; administrated. The reference ranges of the liver blood test were <30 for AST, <30 for ALT, 100–325 for ALP, and 0.2–1.2 for T. Bil. The threshold numbers of patients in “Suspected drugs” and “ATC level 2 of suspected drugs” were judged by the sum of all types of drug-induced liver injury (DILI).

**Table 2 metabolites-10-00355-t002:** Identified lipid classes and numbers of individual lipids.

Category	Class (Abbreviation)	Class	Number of Lipids
Phosphoglycerolipid	LPC	Lysophosphatidylcholine	12
Phosphoglycerolipid	LPCe	Ether-type lysophosphatidylcholine	2
Phosphoglycerolipid	LPE	Lysophosphatidylethanolamine	5
Phosphoglycerolipid	LPEe	Ether-type lysophosphatidylethanolamine	1
Phosphoglycerolipid	LPI	Lysophosphaidylinositol	2
Phosphoglycerolipid	PC	Phosphatidylcholine	56
Phosphoglycerolipid	PC+O	Oxidized phosphatidylcholine	2
Phosphoglycerolipid	ether-linked PC	Ether-type phosphatidylcholine	40
Phosphoglycerolipid	PE	Phosphatidylethanolamine	15
Phosphoglycerolipid	ether-linked PE	Ether-type phosphatidylethanolamine	29
Phosphoglycerolipid	PI	Phosphatidylinositol	18
Phosphoglycerolipid	PS	Phosphatidylserine	2
Sphingolipid	Cer	Ceramide	14
Sphingolipid	CerG1	Monoglycosylceramide	6
Sphingolipid	CerG1+O	Oxidized monoglycosylceramide	3
Sphingolipid	CerG2	Diglycosylceramide	4
Sphingolipid	CerG3	Triglycosylceramide	4
Sphingolipid	Gb4	Ganglioside Gb4	1
Sphingolipid	GM3	Ganglioside GM3	7
Sphingolipid	GM3+O	Oxidized ganglioside GM3	1
Sphingolipid	SM	Sphingomyelin	37
Sphingolipid	SM+O	Oxidized sphingomyelin	2
Sphingolipid	Su1G1	Sulfatide	1
Neutral lipid	ChE	Cholesterolester	19
Neutral lipid	DG	Diacylglycerol	22
Neutral lipid	TG	Triacylglycerol	138
Other lipid	Car	Acylcarnitine	6
Other lipid	CoQ	CoenzymeQ	1
Other lipid	FA	Fatty acid	5
Other lipid	FAA	Fatty amide	5
Other lipid	Other	Other	3

**Table 3 metabolites-10-00355-t003:** Specific lipids altered in mixed-type DILI.

		vs. Recovered Phase			vs. Healthy M1			vs. Healthy M2			vs. Healthy F1			vs. Healthy F2		
Metabolite ID	Name	*p* Value	Effect Size	ROC–AUC	*p* Value	Effect Size	ROC–AUC	*p* Value	Effect Size	ROC–AUC	*p* Value	Effect Size	ROC–AUC	*p* Value	Effect Size	ROC–AUC
M008	LPC(18:2)	**1.07** × 10^−2^	**−0.91**	0.75	**4.09** × 10^−3^	**−1.17**	**0.81**	**3.85** × 10^−5^	**−1.85**	**0.91**	**2.94** × 10^−2^	**−0.89**	0.72	**3.06** × 10^−2^	**−0.85**	0.72
M014	LPC(16:1e)	**6.48** × 10^−3^	**−0.94**	0.78	**3.40** × 10^−4^	**−1.6**	**0.89**	**3.16** × 10^−5^	**−2**	**0.9**	**1.24** × 10^−2^	**−1**	0.79	**8.68** × 10^−4^	**−1.4**	**0.85**
M104	PC(38:6e)	**1.74** × 10^−2^	**−0.93**	0.78	8.23 × 10^−2^	−0.67	0.71	**3.46** × 10^−2^	**−0.82**	0.72	**2.84** × 10^−2^	**−0.86**	0.75	**3.72** × 10^−3^	**−1.2**	0.78
M140	PE(36:4e)	**1.52** × 10^−2^	**−0.85**	0.76	2.70 × 10^−1^	−0.43	0.69	3.60 × 10^−1^	−0.36	0.66	**5.05** × 10^−3^	**−1.14**	**0.84**	2.13 × 10^−1^	−0.49	0.74
M146	PE(38:4e)	**1.74** × 10^−3^	**−1.22**	**0.82**	2.74 × 10^−1^	−0.4	0.56	1.00 × 10^−1^	−0.62	0.67	**2.60** × 10^−3^	**−1.24**	**0.82**	**3.24** × 10^−2^	**−0.84**	0.73
M151	PE(38:6e)	**1.61** × 10^−2^	**−0.85**	0.75	**2.21** × 10^−2^	**−0.9**	0.73	**6.88** × 10^−3^	**−1.09**	0.76	**5.98** × 10^−3^	**−1.14**	0.79	**1.60** × 10^−2^	**−0.96**	0.75
M185	Cer(d34:1)	**9.02** × 10^−4^	**1.68**	**0.87**	**1.21** × 10^−4^	**2.08**	**0.93**	**7.00** × 10^−4^	**1.69**	**0.87**	**1.33** × 10^−4^	**2.06**	**0.93**	**1.18** × 10^−3^	**1.6**	**0.85**
M186	Cer(d36:1)	**2.90** × 10^−3^	**1.26**	**0.86**	**2.62** × 10^−3^	**1.42**	**0.88**	**5.66** × 10^−3^	**1.26**	**0.85**	**8.77** × 10^−4^	**1.71**	**0.96**	**5.91** × 10^−3^	**1.29**	**0.85**
M224	GM3(d34:1)+O	**1.37** × 10^−3^	**1.22**	**0.83**	**9.78** × 10^−5^	**2.23**	**0.94**	**2.30** × 10^−4^	**1.97**	**0.91**	**6.28** × 10^−4^	**1.75**	**0.89**	**2.33** × 10^−3^	**1.49**	**0.81**

The values fulfilled the threshold values (*p* < 0.05, effect size > 0.8, ROC–AUC > 0.8) are indicated by bold fonts. M1; middle-age male, M2; old-age male, F1; middle-age female, F2; old-age female, where middle age was approximately 45 years and old age was approximately 60 years, ROC: receiver operating characteristics, AUC: area under the curve.

**Table 4 metabolites-10-00355-t004:** Specific lipids altered in cholestatic-type DILI.

		vs. Recovered Phase			vs. Healthy M1			vs. Healthy M2			vs. Healthy F1			vs. Healthy F2		
Metabolite ID	Name	*p* Value	Effect Size	ROC–AUC	*p* value	Effect SIZE	ROC–AUC	*p* Value	Effect Size	ROC–AUC	*p* Value	Effect Size	ROC-AUC	*p* Value	Effect Size	ROC–AUC
M023	PC(30:0)	**2.68** × 10^−2^	**0.89**	0.72	**3.05** × 10^−3^	**2.59**	**0.99**	**5.61** × 10^−3^	**2.23**	**0.96**	1.54 × 10^−2^	**1.66**	**0.89**	**3.39** × 10^−2^	**1.31**	**0.82**
M024	PC(31:0)	**2.92** × 10^−2^	**1.31**	**0.84**	**5.89** × 10^−4^	**3.16**	**0.99**	**6.21** × 10^−4^	**3.06**	**0.98**	3.25 × 10^−3^	**2.16**	**0.93**	**6.31** × 10^−3^	**1.82**	**0.88**
M026	PC(32:1)	**1.90** × 10^−2^	**0.91**	0.72	**7.82** × 10^−3^	**2.23**	**0.98**	**1.10** × 10^−2^	**2.05**	**0.94**	1.86 × 10^−2^	**1.74**	**0.87**	**2.73** × 10^−2^	**1.57**	**0.83**
M028	PC(33:1)	**2.60** × 10^−2^	**1.36**	**0.88**	**1.29** × 10^−3^	**2.37**	**0.96**	**1.44** × 10^−3^	**2.88**	**0.99**	4.99 × 10^−3^	**2.05**	**0.94**	**9.55** × 10^−3^	**1.58**	**0.88**
M096	PC(36:5e)	**4.36** × 10^−2^	**−0.8**	0.69	**1.27** × 10^−2^	**−1.45**	**0.83**	**8.67** × 10^−3^	**−1.14**	**0.8**	**6.55** × 10^−3^	**−1.45**	**0.86**	**2.30** × 10^−2^	**−1.24**	0.77
M104	PC(38:6e)	**1.45** × 10^−3^	**−0.92**	0.73	5.38 × 10^−2^	**−1.06**	0.78	**2.23** × 10^−2^	**−1.09**	**0.81**	**2.41** × 10^−2^	**−1.16**	0.78	**1.46** × 10^−2^	**-1.49**	**0.83**
M140	PE(36:4e)	**9.57** × 10^−3^	**−1.65**	**0.86**	7.10 × 10^−2^	−0.78	0.73	1.03 × 10^−1^	−0.7	0.74	**1.18** × 10^−3^	**−1.39**	**0.88**	5.27 × 10^−2^	**−0.85**	0.76
M146	PE(38:4e)	**2.68** × 10^−2^	**−1.55**	**0.89**	9.48 × 10^−2^	−0.62	0.68	**2.93** × 10^−2^	**−0.85**	0.78	**8.03** × 10^−4^	**−1.38**	**0.91**	**9.08** × 10^−3^	**−1.2**	**0.84**
M157	PE(40:6e)	**4.92** × 10^−3^	**−1.66**	**0.89**	6.09 × 10^−2^	−0.69	0.72	**8.24** × 10^−3^	**−0.99**	**0.88**	**6.23** × 10^−5^	**−1.83**	**0.95**	**2.43** × 10^−4^	**−1.61**	**0.98**
M160	PE(40:7e)	**2.45** × 10^−3^	**−1.73**	**0.91**	**4.44** × 10^−2^	−0.76	0.74	**2.30** × 10^−4^	**−1.63**	**0.96**	**8.09** × 10^−5^	**−1.73**	**0.93**	**2.99** × 10^−4^	**−1.65**	**0.94**
M161	PE(40:7e)	**1.11** × 10^−2^	**−0.8**	0.75	3.58 × 10^−1^	−0.42	0.61	**1.12** × 10^−2^	**−1.24**	**0.81**	**4.52** × 10^−2^	**−0.98**	0.73	**6.81** × 10^−3^	**−1.44**	**0.84**
M174	PI(38:3)	**8.80** × 10^−3^	**−0.96**	**0.83**	**2.36** × 10^−2^	**−1.18**	**0.83**	**2.72** × 10^−2^	**−1.14**	**0.82**	**2.15** × 10^−2^	**−1.17**	**0.82**	**2.28** × 10^−2^	**−1.14**	**0.83**
M175	PI(38:4)	**7.65** × 10^−3^	**−1.18**	**0.83**	**1.24** × 10^−3^	**−1.85**	**0.9**	**2.65** × 10^−4^	**−2.2**	**0.95**	**2.56** × 10^−3^	**−1.68**	**0.88**	**3.88** × 10^−3^	**−1.54**	**0.86**
M179	PI(40:4)	**1.90** × 10^−3^	**−1.15**	**0.82**	1.19 × 10^−1^	**-0.86**	0.73	5.49 × 10^−2^	**−1.07**	0.74	**9.32** × 10^−3^	**−1.41**	0.78	7.07 × 10^−2^	**−1.09**	0.71
M213	CerG3(d40:1)	**1.38** × 10^−2^	**−1.19**	**0.81**	7.12 × 10^−2^	−0.69	0.7	**2.04** × 10^−3^	**−1.26**	**0.86**	**4.20** × 10^−3^	**−1.19**	**0.84**	**3.09** × 10^−5^	**−1.9**	**0.96**
M214	CerG3(d42:1)	**8.26** × 10^−4^	**−1.68**	**0.86**	5.45 × 10^−2^	−0.78	0.72	**2.75** × 10^−3^	**−1.17**	**0.88**	**8.50** × 10^−3^	**−1.06**	**0.81**	**2.85** × 10^−6^	**−2.39**	**0.98**
M215	CerG3(d42:2)	**1.34** × 10^−2^	**−0.82**	0.69	6.34 × 10^−2^	**0.88**	0.78	9.13 × 10^−1^	0.04	0.58	4.44 × 10^−1^	0.34	0.62	2.92 × 10^−1^	−0.53	0.61
M243	SM(d40:1)	**2.86** × 10^−2^	**−0.84**	0.77	**6.17** × 10^−3^	**−1.82**	**0.89**	**8.18** × 10^−4^	**−2.33**	**0.97**	**8.43** × 10^−3^	**−1.77**	**0.89**	**7.56** × 10^−4^	**−1.95**	**0.97**
M316	TG(44:0)	**2.79** × 10^−2^	**−0.84**	0.78	4.52E-01	0.38	0.67	3.30E-01	0.56	0.62	2.11 × 10^−1^	0.79	0.68	8.15 × 10^−1^	0.11	0.5
M450	CoQ10	**7.22** × 10^−4^	**−1.04**	**0.8**	**5.17** × 10^−3^	**−1.55**	**0.85**	**1.79** × 10^−3^	**−1.96**	**0.92**	5.59 × 10^−2^	**−1.23**	**0.82**	1.05 × 10^−1^	**−0.81**	0.78

The values fulfilled the threshold values (*p* < 0.05, effect size > 0.8, ROC–AUC > 0.8) are indicated by bold fonts. M1; middle-age male, M2; old-age male, F1; middle-age female, F2; old-age female, where middle age was approximately 45 years and old age was approximately 60 years, ROC: receiver operating characteristics, AUC: area under the curve.

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
