# Peer review of "Plasma Lipid Profiling of Three Types of Drug-Induced Liver Injury in Japanese Patients: A Preliminary Study"

_metabolites, 2020, doi:10.3390/metabo10090355_

Round 1

Reviewer 1 Report

The submitted manuscript by Saito et al describes the use of an untargeted lipidomics approach to determine changes in the plasma lipid profiles of patients with drug-induced liver injury (DILI). Results are presented that show that the concentrations of molecular species of ceramides and arachidonic acid containing ether-linked phospholipids are altered in mixed type-DILI. Disturbances in the levels of saturated or monounsaturated phospholipids together with arachidonic acid and DHA containing phospholipids in cholestatic-type DILI are also detailed. There are a number of points, which the authors should address:

Specific Comments

Page 1. Title. The title of the manuscript could be modified to ‘Plasma lipid profiling of three types of drug-induced liver injury in Japanese patients’.

Pages 2-3. Results - Table 1. It would be useful to indicate the liver blood test (AST, ALT and bilirubin) references ranges for healthy individuals.

Page 4. Results. As this study was focused on hepatotoxic conditions, were any bile acids (free or conjugated) detected in the lipidomics screen? This point should be discussed in the text.

Page 4. Results. It would be useful to present exemplar LC-MS traces of plasma lipid profiles from healthy individuals and patients with hepatocellular, mixed and cholestatic type DILI. These could be included in the supplementary materials.

Pages 7 and 10. Figures 2 and 3. The y-axes on the graphs state 'lipid levels'. This term should be clarified and the units denoted.

Pages 7-10. Results. It would be interesting to see if the ratios of plasma concentrations of the altered lipid species offer any additional discrimination between DILI patients and healthy subjects and/or the acute and recovery phases of DILI.

Page 11. Discussion. It would be useful if the authors could comment on whether the altered concentrations of arachidonic acid and DHA containing phospholipids might influence the production of pro-inflammatory and pro-resolving lipid mediators in DILI.

Page 11. Discussion. In relation to the changes of specific lipids acting as potential biomarkers for DILI the authors may wish discuss how other disease states and external factors could alter the levels of these lipids.

Page 13. Materials and Methods. Whilst appropriate references are cited in the text, it would be useful to provide experimental details on the lipidomic analyses and associated data processing. This could be included in the supplementary materials.

Page 13. Materials and Methods. It is stated that the quantified raw data were normalized to the internal standard and that following normalization, the median value of each lipid in all samples was set to 1. Further details as to why this was done should be provided.

Reviewer 2 Report

The authors present a generally well written description of a lipidomics study into DILI, an important adverse drug reaction. They used untargeted lipidomics to compare lipid profiles between acute and recovery time points for patients with hepatocellular, mixed, and cholestatic DILI types. Lipid profiles at both time points were also compared to healthy controls divided into groups based on age and sex. The study team identified a number of lipids that were able to differentiate acute DILI from recover and healthy control groups, and these results varied based on DILI subtype.  This is an interesting study but would benefit from some editing/clarifications, listed below:

-Lines 35-36: When you say that mixed and cholestatic types demonstrated specific lipid alterations of plasma, what was the comparator- are you referring to acute vs recovery or acute vs recovery and healthy controls? Please clarify

-Table 1: It is not clear what is going on with the all caps drug classes until you get into the text beneath the table. This should be explained in the table footnotes. It is also unclear what is going on with the suspect drugs. In column 1 there is a note that the meds were present in more than 2 cases, but in several columns the number of cases is listed as 1.

-Table 3: Define F1, F2, M1, and M2 within the table of the table footnotes. These abbreviations were not clear until text beneath the table was read.

-Tables 3 and 4: Could these be condensed so that metabolite ID and Name are not repeated for every comparison group (the healthy groups and recovery phases with their p values, effect size, and ROC-AUC as unique columns within the same table)? I think this would make it easier to compare changes across the different groups.

-First sentence in discussion: when you say that specific lipid alterations in the plasma occur for 2 of the 3 types of DILI, state your comparator (vs healthy, acute vs recovered, etc- as suggested in the abstract comment for lines 35-36).

-Why are the p-values not adjusted for multiple comparisons?

-It is interesting that you separated the healthy controls into groups based on sex and age, but this isn't addressed in the discussion.

Reviewer 3 Report

Manuscript: Plasma lipid profiling of three types of drug-induced liver injury in Japanese

Feedback for the author(s)

The authors in this work carried out a lipidomics analysis on plasma from patients with drug-delivered liver injury (DILI) in order to identify new biomarkers. The comparison of lipids levels between the acute and recovery phase of hepatocellular, mixed and cholestatic types of DILI patients revealed specific lipid alterations only in mixed and cholestatic types. Moreover, the authors showed that increased levels of several ceramides and decreased levels of arachidonic acid-containing ether-linked phosphoglycerolipids in the acute phase are specific indicators of mixed-type DILI, whereas increased levels of palmitic acid-containing phosphatidylcholines and decreased levels of arachidonic acid- or docosahexaenoic acid-containing ether-linked phosphoglycerolipids and phosphatidylinositols are specific markers of cholestatic-type DILI. Finally, they identified lipids well capable of discriminating the acute phase from the recovery phase, thus providing a contribution to the identification of candidate biomarkers of DILI. More interestingly, the authors suggested the use of some phosphoglycerolipids as possible therapeutic agents for mixed- and cholestatic-type DILI treatment and the target of ceramides to pharmacologically counteract inflammation. I think it is a well-structured work, clearly designed from an experimental point of view and faithfully discussed through an elegant summary of the main evidences. However, I think that the authors need to be clarify some aspects in order to make the paper suitable for publication:

  1. In line 291 the authors state: “Fourth, we did not control the alcohol and food intake of the patients and the time of blood draw was not standardized, both of which might have affected plasma lipid levels”. I suggest to explicit this concept because at the first reading I understood that the authors don’t know if patients were fasted before sample collection. This is a not acceptable condition for a plasma lipidomics study. The authors should report in the method section at least the "fasting condition" before blood collection of the clinical cohort analysed.

  1. I would suggest changing the title to Plasma lipid profiling of three types of drug-induced liver injury in Japanese patients: a preliminary study

  1. To emphasize the values in the tables where the specific lipids altered are listed, it seems more than enough to put them in bold. Underlying them on top of that make the tables less clear.

  1. Table 3 and 4 report the value of AUC obtained by ROC curve from each differential compound in the different analysed condition. It would be interesting to see a cumulative ROC curve including the most significant differential lipids for each condition analysed.

  1. I would suggest to briefly describe the CIOMS/RUCAM and DDW-J2004 scoring scales that the authors used during the collection of clinical information of patients (Table 1) in order to establish the causal relationship between suspected drug and liver damage.

  1. Are there significant correlations between the discriminant lipid found and the clinical parameters of the enrolled patients? The authors should discuss this aspect especially in relation to the most important diagnostic parameters reported in Table 1.

  1. I would suggest putting the tables listing the specific lipids altered vs healthy groups after the paragraph (line 151-161) talking about it. I would also suggest to add in the legend of those tables what are HM1, HM2, HF1 and HF2, the same way it is done in Figure 2.

  1. The authors decided to define as altered the lipids that have high “effect size” (|g| > 0.8) and statistically significant differences (p < 0.05) between the acute and recovery phases in each DILI type. The authors should explicit the parameters used to report effect sizes.

  1. In the methods section the authors reported the PC (12:0/12:0) as internal standard. Is this lipid an endogenous lipid in plasma? If yes, which criteria were used to select this specie as IS. Is this PC specie used as IS also for other lipids classes? How was the IS signal used for quantitative processing? The authors should better describe these aspects of the IS used.

  1. The authors evaluated the discrimination ability of altered lipids between the acute and recovery phase in cholestatic- and mixed-type of DILI by ROC analysis, highlighting that some lipids have a bigger areas under the curve, thus showing a better sensitivity and specificity in discriminating. However, the authors should insert a cross-validation to confirm the good predictivity of the proposed models.

  1. The authors admitted they used self-reported healthy subjects as controls without considering possible silent diseases, while promising to update their protocols by future large-scale and standardized studies. However, it would be interesting to insert at least the average of BMI (body mass index) of control patients and the duration of treatment with the drugs, both in DILI patients and controls, if available.

  1. The authors should divide the lipidomics methods from the statistical ones.

  1. At line 75, there might be a missing reference, as the authors declare that “plasma lipidomics of liver phospholipidosis demonstrated increased levels of d18:1/24:0 glucosylceramide (GluCer), which was proposed as a biomarker for the disease”.

  1. At line 76, the sentence should be corrected: “[...] the characterization of overall plasma lipid profiles could lead to a better understanding of hepatic physiology […]”.

  1. At line 107, the word “plasma” is repeated.

  1. At line 111, the authors explain that they use a semicolon “;” to distinguish the stereoisomers. Could they confirm they meant a slash “/”?

  1. At line 158 and 203, there is a space missing “AUC values > 0.8”. At the opposite, at line 228 there is an extra space “PC(33:1; 15:0/18:1, 16:0/17:1)”.

  1. At line 224, “forth” should be replaced by “fourth”.

  1. A general update of the references should be done:
    • The reference 9 is related to “Phospholipids in signal transduction of mesangial cells”: I would suggest choosing another reference that would regard the liver rather than the kidney.
    • Reference 25 declares that “Phosphatidylinositol decreased in most fatty acids” while the opposite is said in the present article: “A previous study demonstrated that microsomal PIs also increased in the liver of a cholestatic rat 243 model”.
    • Reference 37 should be updated because there are novel studies about DILI in Japan between 2010 and 2018.

Round 2

Reviewer 3 Report

Manuscript ID metabolites-885383

Feedback for the author(s)

Authors revised the manuscript taking into consideration the concerns raised in the first revision step. However, I Think that some point from the first step of revision have not been well justified and the authors at least need to report their justification explicitly in the paper. In particular I referred to the following points:

Specific Comment  1° revision:

  1. In line 291 the authors state: “Fourth, we did not control the alcohol and food intake of the patients and the time of blood draw was not standardized, both of which might have affected plasma lipid levels”. I suggest to explicit this concept because at the first reading I understood that the authors don’t know if patients were fasted before sample collection. This is a not acceptable condition for a plasma lipidomics study. The authors should report in the method section at least the "fasting condition" before blood collection of the clinical cohort analysed.

Authors Response: We understand the concern of the Reviewer #3 on the effects of with or without fasting on the lipidomics data. However, recent literature has demonstrated that the global impact of postprandial effect on lipidomics is less than the inter-individual variations even with the selected age range and sex (Saito et al., 2020 10:185). In addition, it is not only ethically but also practically difficult to control the food intake of the subject of DILI patients especially on the acute phase. Therefore, we believe that the unknown state of fasting can be acceptable for our present study, although the impact of post-prandial effect should be considered. Inconsideration of the concern, we added “a preliminary study” to the title as Reviewer #3 suggested in the Comment 2

Comment to the authors by 2° revision:

The authors declare that “recent literature demonstrated that the global impact of postprandial effect on lipidomics is less than the inter-individual variations even with the selected age range and sex (Saito et al., 2020 10:185)”.

The paper mentioned is of the same first author of this work and reported the low impact of postprandial effect on lipidomics in CSF analysis, therefore it shouldn’t be cited for this concern.

 I don’t read all the manuscript mentioned, but I suppose that CSF lipidomics may be few influenced by fasting or not fasting condition. I can’t believe that plasma lipidome is not influenced by fasting/not fasting condition. Moreover, there are a lot of papers demonstrating a post-prandial influence in plasma lipidomics studies (es. Takeo Moriya et al. J Biochem.2018 Feb 1;163(2):113-121. doi: 10.1093/jb/mvx066.)

Thus, I suggest the authors to report explicitly this limitation in the text, for example:

“In addition, it is practically difficult to control the food intake of the subject of DILI patients especially on the acute phase. Therefore, we believe that the unknown state of fasting can be acceptable for our present preliminary study, although the impact of post-prandial effect should be considered”.

Specific comment  1° revision:

  1. In the methods section the authors reported the PC (12:0/12:0) as internal standard. Is this lipid an endogenous lipid in plasma? If yes, which criteria were used to select this specie as IS. Is this PC specie used as IS also for other lipids classes? How was the IS signal used for quantitative processing? The authors should better describe these aspects of the IS used.

Authors Response: Endogenous PC (12:0/12:0) is not detectable in our lipidomics platform. We added this description in sub-section of “Lipidomics” in the Materials and Methods section. We used IS for normalization of other lipid classes, which is one of the general approaches to normalize lipidomics data when present lipid levels have an arbitrary unit.

Comment to the authors by 2° revision:

The authors should discuss their choice in the selection of one IS and indicate limitation in the quantification of different classes of lipids by using a single IS (see for example: “Miao Wang et al. Mass Spectrom Rev 2017 Nov;36(6):693-714. doi: 10.1002/mas.21492.”).
